# An Effectiveness Evaluation of Nucleo-Annuloplasty for Lumbar Discogenic Lesions Using Disc-FX: A Scoping Review

**DOI:** 10.3390/medicina59071291

**Published:** 2023-07-13

**Authors:** Guang-Xun Lin, Shang-Wun Jhang, Chien-Min Chen

**Affiliations:** 1Department of Orthopedics, The First Affiliated Hospital of Xiamen University, School of Medicine, Xiamen University, Xiamen 361000, China; 2The School of Clinical Medicine, Fujian Medical University, Fuzhou 350122, China; 3Division of Neurosurgery, Department of Surgery, Changhua Christian Hospital, Changhua 500209, Taiwan; 4Department of Leisure Industry Management, National Chin-Yi University of Technology, Taichung 41170, Taiwan

**Keywords:** degenerative disk disease, diskogenic pain, Disc-FX, radiofrequency, nucleo-annuloplasty

## Abstract

*Background and Objectives*: Degenerative disk disease is a widespread chronic condition that causes diskogenic pain. Diskogenic pain can be treated with various therapy methods. Disc-FX is a revolutionary, minimally invasive, percutaneous nucleo-annuloplasty method that combines manual diskectomy with nuclear and annular remodeling using radiofrequency ablation to relieve diskogenic pain. In this study, the technical features, clinical outcomes, and complications of Disc-FX are summarized. *Materials and Methods*: A comprehensive literature review was performed. By exploring several databases, we collected studies on Disc-FX for treating diskogenic pain. The outcomes included perioperative data, clinical results, and complications. *Results*: In the 15 studies included, data from 570 patients were collected. L4–L5 was the most frequently operated level, and most cases underwent single-level procedures. The follow-up period for these patients ranged from 2 months to 24 months. One study reported a procedure time between 35 and 60 min, whereas the remaining studies reported a procedure time of less than 30 min. The mean visual analog scale score decreased from 7.22 preoperatively to 1.81 at the final follow-up. The mean numerical rating scale score decreased from 6.98 preoperatively to 3.9 at the final follow-up. The mean Japanese Orthopaedic Association score improved from 16.26 preoperatively to 25.88 in the final follow-up. The mean Oswestry Disability Index score decreased from 35.37 preoperatively to 14.66 at the final follow-up. The mean satisfaction rate (based on the Macnab criteria) was 87.6% (range, 78.4–95.2%). The total incidence of postoperative transient pain was 8.77% (50/570) after nucleo-annuloplasty using Disc-FX, and recurrence was 1.58% (9/570). *Conclusions*: According to our comprehensive evaluation, using percutaneous nucleo-annuloplasty for treating lumbar diskogenic diseases provided considerable pain alleviation and improved functional outcomes with fewer complications. Disc-FX is a safe and effective procedure that is a good treatment option for patients with diskogenic pain.

## 1. Introduction

In the past, individuals with degenerative disk disease or lumbar disk herniation resulting in diskogenic pain were treated conservatively with rest, medications, injections, and/or physical therapy [1,2,3]. Unfortunately, these treatments have not always been effective. Those for whom previous conservative treatments have not been effective are forced to live with their symptoms or consider spine surgery [4]. If these patients undergo traditional spinal decompression surgery, they may require weeks or months of recuperation, which will have a greater impact on their daily lives and work.

Minimally invasive approaches for treating degenerative disk diseases have several advantages, including greater preservation of the spine architecture, less tissue loss, and decreased risk [5,6]. In recent years, it has been demonstrated that spinal degeneration is not an isolated process, but is closely related to disc degeneration, endplate degeneration, facet joint degeneration, and fatty infiltration/atrophy of the paraspinal muscles. Therefore, the use of a minimally invasive approach is not just for the protection of one structure, but for the long-term protection of all structures [7,8]. However, controversy exists regarding the surgical indications for diskogenic pain. There may be no need for traditional microsurgical diskectomy [9,10], and the use of full endoscopic lumbar diskectomy may also be problematic [11,12]. With the progressive development of minimally invasive surgical techniques and instrumentation, the Disc-FX (Elliquence, LLC, Baldwin, NY, USA) system of fluoroscopically guided nucleus pulposus removal, radiofrequency ablation, and annuloplasty is gradually being introduced in clinical practice as a new promising approach to treating diskogenic pain and contained-type herniated nucleus pulposus [13]. Disc-FX is the option of choice for those who have exhausted conservative treatment options but are not yet ready for major spine surgery [14]. It is an outpatient procedure where the patient can go home the same day.

The results of several studies suggest that nucleo-annuloplasty using the Disc-FX system may be a reasonable treatment option for those with degenerative disk diseases [13,15]. However, no robust study has demonstrated the efficacy and outcomes of Disc-FX, particularly long-term results. Therefore, this study was designed to systematically review the literature to present a summary of the clinical outcomes and complications related to this technique in degenerative disk diseases resulting in diskogenic pain.

## 2. Materials and Methods

### 2.1. Literature Search

This systematic review followed the Preferred Reporting Items for Systematic Reviews and Meta-Analyses (PRISMA) criteria. From database inception to 15 May 2022, papers published in English or Chinese were searched for from the Cochrane Library, PubMed, Web of Science, China National Knowledge Internet, and Wanfang Data. The following keyword combinations were used to obtain the highest search sensitivity: “Disc-FX”, “nucleo-annuloplasty”, or “intradiscal radiofrequency”. Furthermore, we discovered pertinent papers from references to help with our search. Two researchers separately evaluated the titles and abstracts of all search results. The relevance of those whose material looked to be relevant was then evaluated further. Discussions with a third party were used to settle any disagreements. To prevent any possible data duplication, when analyzing studies from the same institution or authors, we only evaluated the most recent study or the study with a longer follow-up.

### 2.2. Data Extraction

All relevant clinical research and original papers were searched for, including prospective, retrospective, and randomized controlled trials. The collected data included the authors, year of publication, type of study, number of patients, level of surgery, follow-up duration, perioperative data, clinical outcomes, and complications. The primary outcomes were visual analog scale (VAS) scores for back and/or leg pain, numerical rating scale (NRS) scores, Japanese Orthopaedic Association (JOA) scores, and Oswestry Disability Index (ODI) scores, which were measured preoperatively and in the final follow-up. Additionally, Macnab’s outcome evaluation of patient satisfaction was performed. All data were summaries of qualitative research. No statistical analysis or meta-analysis was performed.

### 2.3. Quality Assessment

The Newcastle–Ottawa Scale (NOS) was utilized to evaluate the quality of non-randomized trials. Each study was assessed on the basis of selection, comparability, and exposure/outcome. Employing these criteria, we regarded papers that received five or more stars in our review.

### 2.4. Surgical Technique

All procedures were performed under local anesthesia in the operating room. On the Jackson table, each patient was placed in the prone position. Magnetic resonance imaging (MRI) was used to plan skin entrance locations before surgery. A standard posterolateral approach was used. Under fluoroscopic assistance, the skin entrance location was indicated. The pathological side was the chosen approach side. In the lateral view, the puncture needle was inserted through the Kambin’s triangle at an angle of 30–45° to the dorsal plane, with the needle tip positioned in the posterior third of the intervertebral disk. The needle core was withdrawn, and diskography was performed by injecting a mixture of iontophoresis and melanin in a 9:1 ratio. A guidewire was placed through the puncture needle, and a 0.7 cm incision was made in the skin. Under fluoroscopic guidance, a cannula was introduced through the guidewire over the annular (in anteroposterior and lateral views). A pair of gripping forceps was used for manual diskectomy, and diseased disks were removed separately. The Surgi-Max system (Elliquence, LLC, Baldwin, NY, USA) can control the radiofrequency-generating device using a unique design of bipolar radiofrequency electrodes that can be flexed (Trigger-Flex), extended, and steered to enter the nucleus pulposus through the working channel. First, a radiofrequency of 1.7 MHz was used to ablate the nucleus pulposus with the Bipolar Turbo mode, once for 3–4 s, repeated 4–6 times; then, the Trigger-Flex tip was bent to the dorsal side; we fluoroscopically determined that the Trigger-Flex tip is located in the contralateral fibrous ring, and we slowly pulled out the fibrous ring to the ipsilateral side with 1.7 MHz (Bipolar Hemo mode), once for 4–6 s, which was repeated 4–6 times. The intervertebral space was repeatedly irrigated with saline through the working channel until the irrigation solution was clear. The working channel was removed, without the need for suturing the incision; then, the band-aid patch was applied. The patients were usually dismissed the same or the next day.

## 3. Results

### 3.1. Selection of Studies and Quality Evaluation

Through the database search, 96 studies were discovered. Following a title and abstract screening, which eliminated 5 irrelevant articles, the remaining 17 possibly relevant articles were obtained. Following a thorough examination of the complete text, 15 articles were finally included in this review [13,16,17,18,19,20,21,22,23,24,25,26,27,28,29]. Figure 1 shows a flowchart for a specific study. Appendix A shows the PRISMA checklist.

All 15 studies used a retrospective comparative cohort design with moderate to high quality as assessed by the NOS (Table 1).

### 3.2. Study Characteristics and Outcomes

Table 2 presents the selected study characteristics. Of these, 1 was a prospective study and the remaining 14 were retrospective studies. The Disc-FX procedure was performed in 570 patients. The L4–L5 was the most commonly operated level, and most surgeries were single-level surgeries. The follow-up period for these patients ranged from 2 months to 24 months.

Eight papers (Table 3) provided operative times, and all but one paper reported operative times ranging from 35 to 60 min, whereas the remaining paper reported an operative time of less than 30 min. Only five studies reported estimated blood loss, none of which exceeded 10 mL.

Ten studies (Table 3) provided VAS scores. The mean VAS scores decreased from 7.22 preoperatively to 1.81 at the final follow-up. Two studies provided VAS scores for back pain. The mean VAS scores for back pain decreased from 4.84 preoperatively to 2.02 at the final follow-up. Two studies provided VAS scores for leg pain. The mean VAS scores for leg pain decreased from 3.84 preoperatively to 0.96 at the final follow-up. Three studies provided NRS scores. The mean NRS scores decreased from 6.98 preoperatively to 3.9 at the final follow-up (Table 3).

Five studies provided JOA scores. The mean JOA scores improved from 16.26 preoperatively to 25.88 at the final follow-up (Table 3). Seven studies provided ODI scores. The mean ODI scores decreased from 35.37 preoperatively to 14.66 at the final follow-up (Table 3). The mean satisfaction rate (based on the Macnab criteria) was 87.6% (range, 78.4–95.2%) from eight studies.

### 3.3. Complications

Regarding postoperative complications, three papers reported postoperative transient pain. The total incidence of postoperative transient pain was 8.77% (50/570) after the Disc-FX procedure. The total recurrence rate was 1.58% (9/570) after the Disc-FX procedure reported in four articles. Furthermore, some niche complications have been reported, such as nerve injury (2), hematoma (1), infection (1), lumbar venous plexus injury (1), and reoperation (1).

## 4. Discussion

In this systematic review, we found that postoperative VAS and NRS scores for diskogenic pain using the Disc-FX system were significantly lower than preoperative scores, postoperative ODI scores were lower than preoperative ones, postoperative JOA scores were significantly higher than preoperative scores, and fewer surgical adverse events occurred; additionally, the patient satisfaction rate was 87.6% (range, 78.4–95.2%).

### 4.1. Mechanism and Diagnosis of Diskogenic Pain

Diskogenic pain is discomfort caused by damaged intervertebral disks, and its primary cause is the degeneration and deterioration of the intervertebral disks and the formation of fissures or ruptures in their fibrous rings that separate from each other, stimulating pain receptors distributed on the surface of the fibrous rings of the disks and causing pain [30,31]. Additionally, inflammatory mediators acting on injury receptors at the terminal nerve fibers in the intervertebral disk can cause direct electrophysiological changes or make them extremely sensitive, thus causing pain [32].

Diskogenic pain is difficult to diagnose because it often has only subjective symptoms and lacks objective auxiliary examination indicators. Since patients with diskogenic pain lack obvious manifestations of lumbar disk herniation, locating the painful intervertebral space is often impossible based on symptoms and physical examination, and locating the source of pain using MRI is difficult, whereas diskography can screen normal physiologically degenerated disks from pathologically degenerated disks, thus effectively clarifying the responsible segment and avoiding overtreatment and mistreatment [33,34]. However, the patient’s fear of pain, the intensity of pain during imaging, sensory sensitivities associated with chronic pain disorders, and social factors can contribute to false positives. It can be seen that there is no objective test that can be used independently as a basis for the diagnosis of diskogenic pain; however, it needs to be analyzed and judged along with symptoms and signs. To accurately assess the effectiveness of the Disc-FX system in treating diskogenic pain and contained lumbar disk herniation, a clear diagnosis is necessary.

### 4.2. Features and Advantages of the Disc-FX System

Several conservative treatments have been used for diskogenic pain, including opioids, non-steroidal anti-inflammatory drugs, and physical therapies, such as acupuncture, massage, and physical therapy [35]. Although the use of medication can achieve short-term therapeutic effects, the potential risks of long-term medication are also greatly increased [35]. The failure of physical therapy to achieve good results in treating diskogenic pain in some patients may be related to the complex pathogenesis of the disease. Intradiscal electrotherapy, with its high and difficult-to-regulate temperature, probably causes damage to the surrounding tissues and secondary inflammatory reactions, and its long-term efficacy is uncertain [36]. A sinuvertebral nerve block is a rapid and precise intervention performed under local anesthesia for the treatment of diskogenic pain. A recent study [37] reported on 32 patients with diskogenic pain who underwent a sinuvertebral nerve block. The patients’ VSA and ODI scores decreased significantly immediately after surgery, from 5.75 and 32.59 preoperatively to 2.5 and 17.28 on postoperative day 3, respectively. However, there was a general rebound in the mid- to long-term postoperative period, 3.53 and 19.63 at 1 month, as well as 3.78 and 21.44 at 3 months postoperatively. Eighteen of these patients (56.25%) were observed to experience varying degrees of pain recurrence at 3 months. According to our review, 10 studies provided VAS scores after Disc-FX treatment for diskogenic pain. The mean VAS score decreased from 7.22 preoperatively to 1.81 at the final follow-up. Two studies provided VAS scores for back pain. The mean VAS score for back pain decreased from 4.84 preoperatively to 2.02 at the final follow-up. Two studies provided VAS scores for leg pain. The mean VAS score for leg pain decreased from 3.84 preoperatively to 0.96 at the final follow-up. Seven studies provided ODI scores. The mean ODI score decreased from 35.37 preoperatively to 14.66 at the final follow-up. It is evident that Disc-FX greatly outperforms sinuvertebral nerve block in terms of long-term outcomes for treating individuals with diskogenic pain. In addition, nucleus pulposus removal alone cannot completely remove inflammatory factors and inflammatory tissue from the disc, so a combination of different minimally invasive approaches may have a greater therapeutic effect.

The Disc-FX system is an innovative, minimally invasive technique based on the Yeung endoscopy spine system technology developed by Yeung, which is versatile and fills the gap between interventional and open surgery with multiple functions, such as tissue extraction, nucleus pulposus removal, radiofrequency ablation, annuloplasty, irrigation/aspiration, and observation. The Disc-FX system for treating diskogenic pain is closely related to its possible mechanism of occurrence, which can achieve intra-disk decompression, the inactivation of sinus nerve terminals growing into the ruptured annulus fibrosus fissure, and the destruction of inflammatory factors [14]. Intraoperative methylene blue can eliminate the nociceptive allergic reaction caused by local tissue inflammation and can make normal tissue (nerve root) often stain light blue-green, the degenerated nucleus pulposus and inflammatory granulation tissue stain dark blue, the normal nucleus pulposus tissue stain light blue, and the endplate and fiber ring to not stain, which can help the subsequent clamping and distinguish different tissue types. By changing the position and direction of the working channel, the miniature nucleus pulposus clamp can fully eat away the diseased nucleus pulposus and granulation tissue, reduce the pressure in the intervertebral disk, and reduce the irritation to the nerve terminals.

The Disc-FX system employs a radiofrequency of 1.7 MHz from the Elliquence Surgi-Max generator to achieve nucleus ablation (Bipolar Turbo mode) and annular remodeling (Bipolar Hemo mode), respectively [13]. While conventional radiofrequency mainly destroys cells through heat, the Surgi-Max bipolar radiofrequency system uses high frequencies to resonate with cellular fluid, causing cells to rupture from within; the low temperature output makes the bipolar head resistant to adhesion, minimizing tissue trauma and allowing for the maximum preservation of healthy tissue. Finally, repeated irrigation with saline through the working channel can effectively remove chemical pain mediators, toxic metabolites, any byproducts of radiofrequency treatment and nucleus pulposus, and inflammatory tissue debris from the disk; improve the disk microenvironment; and reduce or eliminate pressure and chemically mediated diskogenic pain.

### 4.3. Indications, Contraindications, and Limitations of the Disc-FX System

The Disc-FX system’s 3 mm outer working channel clearly offers less trauma than conventional endoscopic spine surgery and does not require the steep learning curve and lengthy training required for microscopic surgery. However, the careful monitoring of surgical indications remains necessary to achieve optimal results. The indications for surgery using the Disc-FX system should be strictly limited to diskogenic low back pain because of the mild inclusion of disk herniation and fibular ring tears, particularly in young and middle-aged patients with normal or mildly reduced disk height, while severe disk herniation, free nucleus pulposus, and significant nerve root symptoms should be avoided.

Nevertheless, the Disc-FX system also has its limitations. (1) Because of the restriction of the high iliac crest, the working channel cannot be placed in the optimal position at the L5–S1 level in some patients, and for the L2–L3 level and above, the puncture path is closer to the midline to avoid the kidney. (2) Because all surgeries are performed under X-ray guidance, some are blindfolded, and controlling the quantity of nucleus pulposus tissue excised is difficult, particularly for patients of varying ages and degrees of degeneration, and the amount of nucleus pulposus tissue excised frequently varies substantially. (3) The Disc-FX system is indicated in cases with an intact fibrous ring and posterior longitudinal ligament, although it has also applied in cases with fibrous ring tears. (4) Unlike full endoscopic spine surgery, which has multiple surgical approaches, Disc-FX has only one posterolateral approach, limiting its extensive use. (5) Finally, it is not a fully visualized surgical approach.

### 4.4. Management and Prevention of Complications

The following are possible complications associated with the Disc-FX system: (1) Postoperative transient pain is caused by recurrent episodes of transient pain caused by the space formed after the disk is removed and then filled with blood or other tissues causing compression or intraoperative manipulation that irritates the nerve roots and causes edema [38]; this complication is usually relieved by symptomatic treatment and physical therapy. (2) Recurrence may be attributed to the patient having more deteriorated disk tissue that was not removed intraoperatively and the patient having previously engaged in hard physical labor, which impaired the repair of the annulus fibrosus, resulting in recurrence. To avoid recurrence or induced disk herniation in patients with severe intraoperative diskography tears, the degenerated nucleus pulposus should be removed during surgery, and the amount of postoperative activity should be gradual, avoiding strenuous exercise for 3 weeks, and heavy physical exertion should be gradually resumed after surgery. (3) Infections may be related to the patient being elderly, being immunocompromised, or having diabetes, which may improve with intravenous drips of sensitive antibiotics and effective glycemic control. (4) Hematomas, which present as a persistent postoperative exacerbation of symptoms, are relieved via a secondary laminectomy to remove the hematoma. In addition to careful and gentle intraoperative handling, for middle-aged and elderly patients who have been taking aspirin for a long time, the procedure should be performed only after 2 weeks of drug withdrawal. (5) Finally, other rare complications include nerve root injuries and vessel ruptures. These can be prevented by focusing on the patient’s preoperative examination and perioperative management and by strict asepsis and careful manipulation during the procedure.

### 4.5. Strengths and Limitations

This is the first systematic review of the efficacy of nucleo-annuloplasty using Disc-FX for treating lumbar diskogenic lesions. However, this review also has some limitations. This review is of a low evidence-based grade because it consists mainly of retrospective studies; there are no randomized controlled trials or prospective comparative studies to refer to. Additionally, there was variation in the duration of follow-up in this study, which may have provided the incomplete reporting of complication rates. Finally, some studies failed to disclose accurate data on issues such as age, gender, and follow-up duration.

## 5. Conclusions

According to our comprehensive evaluation, percutaneous nucleo-annuloplasty for lumbar diskogenic diseases provided considerable pain alleviation and improved functional outcomes with fewer complications. Disc-FX is a safe and effective procedure that is a good treatment option for patients with diskogenic pain. Existing research, including several that are still underway, are primarily confined to small cohorts and short-term follow-ups. Further prospective studies and randomized trials with large sample sizes and long-term follow-ups should be conducted.

## Figures and Tables

**Figure 1 medicina-59-01291-f001:**
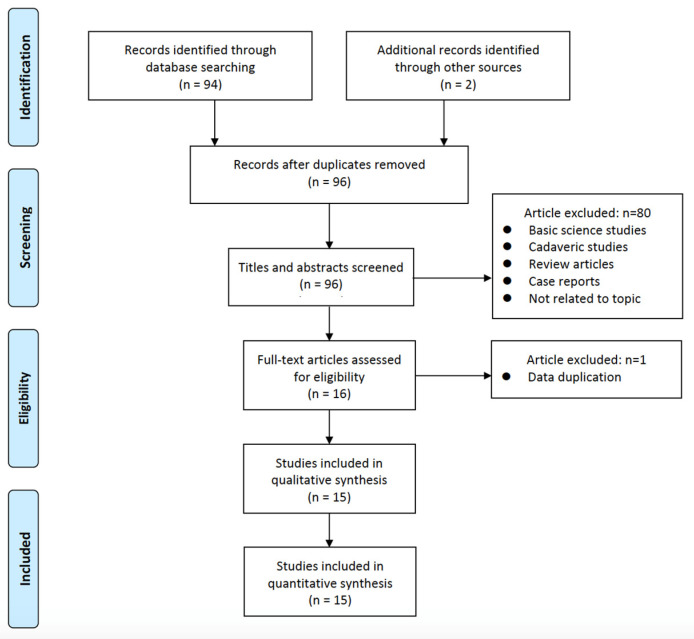
Flowchart of specific studies.

**Table 1 medicina-59-01291-t001:** Quality assessment of the included studies.

Studies	Selection				Comparability	Exposure			Total Scores (of 9)
	Is the Case Definition Adequate?	Representativeness of the Cases	Selection of Controls	Definition of Controls	Comparability of Cases and Controls on the Basis of the Design or Analysis	Ascertainment of Exposure	Same Method of Ascertainment for Cases and Controls	Non-Response Rate	
Bai et al. [16], 2011	**☆**	**☆**	**☆**	**☆**	**☆☆**	**☆**	**☆**		**8** **☆**
Chen et al. [17], 2014	**☆**	**☆**	**☆**	**☆**	**☆**	**☆**	**☆**		**7** **☆**
Hirano et al. [18], 2018	**☆**	**☆**	**☆**	**☆**	**☆**	**☆**	**☆**		**7** **☆**
Kumar et al. [13], 2018	**☆**	**☆**	**☆**	**☆**	**☆☆**	**☆**	**☆**	**☆**	**9** **☆**
Lu et al. [19], 2013	**☆**	**☆**	**☆**	**☆**	**☆**	**☆**	**☆**		**7** **☆**
Liao et al. [20], 2011	**☆**	**☆**	**☆**	**☆**	**☆☆**	**☆**	**☆**		**8** **☆**
Ma et al. [21], 2021	**☆**	**☆**	**☆**	**☆**	**☆☆**	**☆**	**☆**		**8** **☆**
Ou et al. [22], 2013	**☆**	**☆**	**☆**	**☆**	**☆☆**	**☆**	**☆**	**☆**	**9** **☆**
Park et al. [23], 2015	**☆**	**☆**	**☆**	**☆**	**☆☆**	**☆**	**☆**		**8** **☆**
Park et al. [24], 2019	**☆**	**☆**	**☆**	**☆**	**☆☆**	**☆**	**☆**		**8** **☆**
Wang et al. [25], 2013	**☆**	**☆**	**☆**	**☆**	**☆☆**	**☆**	**☆**		**8** **☆**
Xi et al. [26], 2012	**☆**	**☆**	**☆**	**☆**	**☆☆**	**☆**	**☆**	**☆**	**9** **☆**
Yam et al. [27], 2021	**☆**	**☆**	**☆**	**☆**	**☆☆**	**☆**	**☆**	**☆**	**9** **☆**
Zhang et al [28].,2011	**☆**	**☆**	**☆**	**☆**	**☆☆**	**☆**	**☆**		**8** **☆**
Zhang et al. [29], 2015	**☆**	**☆**	**☆**	**☆**	**☆☆**	**☆**	**☆**	**☆**	**9** **☆**

The NOS score evaluates the quality of the literature using the semi-quantitative principle of the star system, except for comparability which can be rated up to 2 stars, the rest of the entries can be rated up to 1 star out of 9 stars, the higher the score indicates the higher quality of the study.

**Table 2 medicina-59-01291-t002:** Characteristics from reviewed studies.

Authors and Year	Study Type	Study Period	No. of Patients (Levels)	Operated Levels	Age (Years), Mean; Range	Sex (M:F)	Follow-Up Period (mo)	Complications (n)
Bai et al. [16], 2011	Retrospective	2010.07–2010.10	36 (NR)	NR	47.5; 18–77	16:20	6	None
Chen et al. [17], 2014	Retrospective	2011.10–2013.02	36 (36)	L3–4 (5); L4–5 (21); L5–S1 (10)	43.5; 26–65	21:15	6–12	None
Hirano et al. [18], 2018	Retrospective	NR	10 (10)	L1–2 (1); L3–4 (2); L4–5 (6); L5–S1 (2)	47.2; 30–72	8:2	6	None
Kumar et al. [13], 2018	Prospective	2010.09–2014.12	51 (66)	L2–3 (1); L3–4 (8); L4–5 (27); L5–S1 (30)	41: 20–63	38:13	24	Infection (1), postoperative transient pain (16)
Lu et al. [19], 2013	Retrospective	2011.02–2012.05	35 (35)	NR	36.5; 27–56	23:12	14.8	Recurrent (1)
Liao et al. [20], 2011	Retrospective	2008.05–2010.05	25 (NR)	NR	NR	NR	12	None
Ma et al. [21], 2021	Retrospective	2013.01–2015.12	56 (NR)	NR	51.2 ± 12.4; 43–65	32:24	3	Recurrent (5), nerve root injury (2); postoperative transient pain (8)
Ou et al. [22], 2013	Retrospective	2010.09–2011.10	62 (70)	L3–4 (5); L4–5 (36); L5–S1 (29)	45.3 ± 16.9: 21–75	25:37	6	None
Park et al. [23], 2015	Retrospective	NR	43 (NR)	NR	44.9; 22–77	30:13	6	None
Park et al. [24], 2019	Retrospective	NR	43 (NR)	NR	56.7 ± 14.1; NR	20:23	6	None
Wang et al. [25], 2013	Retrospective	NR	28 (NR)	NR	NR; 27–73	NR	2	Recurrent (2), lumbar venous plexus injury (1)
Xi et al. [26], 2012	Retrospective	2010.07–2011.06	36 (73)	L2–3 (4); L3–4 (13); L4–5 (32); L5–S1 (24)	56; 18–77	16:20	12	Infection (1), postoperative hematoma (1), postoperative transient pain (26)
Yam et al. [27], 2021	Retrospective	2017–2019	16 (24)	L2–3 (1); L3–4 (3); L4–5 (12); L5–S1 (8)	NR; 23–69	13:3	>6	Re-operation (1)
Zhang et al [28], 2011	Retrospective	2010.02–2011.02	40 (47)	L3–4 (3); L4–5 (23); L5–S1 (21)	38.7; 32–58	22:18	13.8 (6–18)	Recurrent (1)
Zhang et al. [29], 2015	Retrospective	2010.06–2011.05	53 (NR)	NR	NR; 29–56	24:29	24	None

NR: not reported.

**Table 3 medicina-59-01291-t003:** Outcomes from reviewed studies.

Authors and Year	Operative Time (mins)	Blood Loss (mL)	Preoperative		Final Follow-Up		Macnab
			Pain Rating Scale	Functional Rating Scale	Pain Rating Scale	Functional Rating Scale	
Bai et al. [16], 2011	15–20	0–5	VAS: 6.5 ± 1.8	JOA: 19.3 ± 3.6; ODI: 18.7 ± 11.8	VAS: 2.5 ± 2.2	JOA: 25.5 ± 3.2; ODI: 9.1 ± 8.5	81.3%
Chen et al. [17], 2014	15–30	5	VAS: 8.5 ± 1.4	JOA: 12.3 ± 1.2	VAS: 2.1 ± 0.8	JOA: 25.5 ± 2.1	NR
Hirano et al. [18], 2018	NR	NR	VAS: 8.0	JOA: 13.0	VAS: 1.2	JOA: 25.9	NR
Kumar et al. [13], 2018	NR	NR	VAS: 6.69 ± 0.93	ODI: 47.80 ± 17.92	VAS: 2.85 ± 1.76	ODI: 19.63 ± 14.14	78.4%
Lu et al. [19], 2013	29	NR	VAS: 6.70 ± 1.26	ODI: 19.9 ± 6.8	VAS: 1.05 ± 0.66	ODI: 8.6 ± 4.5	NR
Liao et al. [20], 2011	30 ± 5	2 ± 1	VAS: 7.8 ± 0.4	ODI: 39.1 ± 3.9	VAS: 2.0 ± 0.1	ODI: 11.4 ± 1.6	NR
Ma et al. [21], 2021	NR	NR	VAS: 6.2 ± 1.5	JOA: 17.4 ± 3.8	VAS: 2.2 ± 1.3	JOA: 25.9 ± 1.3	89.3%
Ou et al. [22], 2013	25.16 ± 3.21	0–5	VASB: 3.07 ± 1.15; VASL: 6.72 ± 1.26	NR	VASB: 0.98 ± 0.54; VASL: 0.97 ± 0.58	NR	95.2%
Park et al. [23], 2015	NR	NR	NRS: 7.4 ± 0.8	NR	NRS: 3.7 ± 1.9	NR	NR
Park et al. [24], 2019	NR	NR	NRS: 7.3 ± 0.8	ODI: 57.2 ±10.0	NRS: 3.6 ± 1.8	ODI: 22.1 ± 8.4	NR
Wang et al. [25], 2013	NR	NR	VAS: 8.0 ± 1.2	NR	VAS: 1.2 ± 0.8	NR	93%
Xi et al. [26], 2012	18	0–5	VAS: 6.5 ± 1.8	JOA: 19.3 ± 3.6; ODI: 18.7 ± 11.8	VAS: 2.1 ± 1.7	JOA: 26.6 ± 2.4; ODI: 7.7 ± 6.5	78.9%
Yam et al. [27], 2021	NR	NR	NRS: 6.25	ODI: 46.25	NRS: 4.4	ODI: 24.12	NR
Zhang et al [28], 2011	26	<0	VASB: 6.60 ± 1.47; VASL: 0.95 ± 0.63	NR	VASB: 3.05 ± 1.23; VASL: 0.95 ± 0.54	NR	92.5%
Zhang et al. [29], 2015	35–60	NR	VAS: 7.3 ± 1.1	NR	VAS: 0.9 ± 0.2	NR	92.4%

NR: not reported; VAS: visual analog scale; VASB: visual analog scale scores for back pain; VASL: visual analog scale for leg pain; NRS: numerical rating scale scores; JOA: Japanese Orthopaedic Association scores; ODI: Oswestry Disability Index.

## Data Availability

The datasets are presented within the manuscript.

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
