# Peer review of "An Effectiveness Evaluation of Nucleo-Annuloplasty for Lumbar Discogenic Lesions Using Disc-FX: A Scoping Review"

_medicina, 2023, doi:10.3390/medicina59071291_

Round 1

Reviewer 1 Report (Previous Reviewer 2)

Many concerns have been well-addressed.

I think this paper deserves to be accepted.

Author Response

Thank you for your kind comments. We appreciate the recognition of our work.

Reviewer 2 Report (Previous Reviewer 1)

The authors analyzed a minimally invasive method, nucleo-annuloplasty for lumbar discogenic lesions.

In the introduction part, the authors stated that conservative treatments should be conveyed before surgical intervention. Despite minimally invasive surgeries, still the spine and paraspine are affected with those modalities, and surgery should be kept for a second hand intervention if not urgently necessary.

In recent years, it has been demonstrated that spine degeneration is not an isolated process, rather disc degeneration, end-plate degeneration, facet joint degeneration and fatty infiltration/atrophy of paraspinal muscles in close relationship. So using minimally invasive methods are not protective only for one structure rather for all of them in long-term period.

Ref: 

1.     Ekşi MŞ, Özcan-Ekşi EE, Orhun Ö, Turgut VU, Pamir MN: Proposal for a new scoring system for spinal degeneration: Mo-fi-disc. Clin Neurol Neurosurg 2020; 198: 106120.

2.     Teichtahl AJ, Urquhart DM, Wang Y, Wluka AE, Sullivan RO, Jones G, Cicuttini FM: Lumbar disc degeneration is associated with modic change and high paraspinal fat content – a 3.0 T magnetic resonance imaging study. BMC Musculoskeletal Disord. 2016; 17: 439. 

It is very unclear that why the authors described the surgical steps, if the study was a systematic review of the literature. If this is a systematic review, then they should omit the surgical step description. If this is a presentation of the case series of the authors, then why there is no relevant data of their cases in the paper?

It is not clear if the dataset obtained from the literatüre normally distributed or not? If not normally distributed, then median (with interquartile values) values should be given.

The authors gave some numbers of the pooled dataset. However, the change in VAS, ODI, and other parameters should have been presented alltogether with the follow-up periods.

Radiofrequency ablation could lead to fatty degeneration of the paraspinal muscles in long term, hence causing low back pain for another reason, a kind of rebound effect. So how would the authors handle this issue in long term follow-ups of the subjects?

Author Response

The authors analyzed a minimally invasive method, nucleo-annuloplasty for lumbar discogenic lesions.

In the introduction part, the authors stated that conservative treatments should be conveyed before surgical intervention. Despite minimally invasive surgeries, still the spine and paraspine are affected with those modalities, and surgery should be kept for a second-hand intervention if not urgently necessary.

In recent years, it has been demonstrated that spine degeneration is not an isolated process, rather disc degeneration, end-plate degeneration, facet joint degeneration and fatty infiltration/atrophy of paraspinal muscles in close relationship. So, using minimally invasive methods are not protective only for one structure rather for all of them in long-term period.

Ref:

  1. Ekşi MŞ, Özcan-Ekşi EE, Orhun Ö, Turgut VU, Pamir MN: Proposal for a new scoring system for spinal degeneration: Mo-fi-disc. Clin Neurol Neurosurg 2020; 198: 106120.
  2. Teichtahl AJ, Urquhart DM, Wang Y, Wluka AE, Sullivan RO, Jones G, Cicuttini FM: Lumbar disc degeneration is associated with modic change and high paraspinal fat content – a 3.0 T magnetic resonance imaging study. BMC Musculoskeletal Disord. 2016; 17: 439.

 Thank you for your kind comments. Based on your suggestions, we have added the above to the introduction section and cited the references.

It is very unclear that why the authors described the surgical steps, if the study was a systematic review of the literature. If this is a systematic review, then they should omit the surgical step description. If this is a presentation of the case series of the authors, then why there is no relevant data of their cases in the paper?

 Thank you for your kind comments. This paper is a review paper and the reason for describing the surgical techniques is to give the reader a clearer understanding of these techniques.

It is not clear if the dataset obtained from the literature normally distributed or not? If not normally distributed, then median (with interquartile values) values should be given.

 Thank you for your kind comments. I agree with you, but the median and the interquartile values are not given in the original paper, so we can't provide those data in this paper.

The authors gave some numbers of the pooled dataset. However, the change in VAS, ODI, and other parameters should have been presented altogether with the follow-up periods.

 Thank you for your kind comments. According to your suggestions we give the changes in the parameters of the clinical indicators, as shown in Table 3.

Radiofrequency ablation could lead to fatty degeneration of the paraspinal muscles in long term, hence causing low back pain for another reason, a kind of rebound effect. So how would the authors handle this issue in long term follow-ups of the subjects?

 Thank you for your kind comments. As I wrote above about why the surgical technique section was included in the main text, the Disc-FX technique of radiofrequency ablation is performed within the disc space and does not involve damage to the muscles of the low back or fatty degeneration of the paraspinal muscles. Please refer to the surgical technique section for details of the technique.

This manuscript is a resubmission of an earlier submission. The following is a list of the peer review reports and author responses from that submission.

Round 1

Reviewer 1 Report

The authors retrieved 15 papers, one of which was prospective and the others were retrospective. It is not understandable why they did not do quantitative analysis despite they mentioned it in their PRISMA flowchart.

Author Response

Thank you for your kind comments. We added a quality assessment of the included studies as Table 1.

Reviewer 2 Report

I thank the editor for giving me the opportunity to review your paper. Below, I have provided our comments.

Title

Effectiveness evaluation of nucleo-annuloplasty for lumbar discogenic lesions using Disc-FX: A systematic review

Critique

This systematic review deal with the effectiveness of nucleo-annuloplasty for lumbar discogenic lesions using Disc-FX. This system, as minimally invasive approaches, provide many advantages compared to other methods. Therefore, it has archival values but there have some concerns in this study.

1) Introduction: As far as I’m concerned (as reviewer), I think the indication of nucleo-annuloplasty is essential issues in this study. In the background, the indication must be clearly addressed. Where indication were overlapped and many studies have been suggested the effectiveness compared to other surgeries. The systematic reviews were recommended that this point be presented in the background.

2) Materials and Methods: Shouldn’t the order of 2.1 and 2.2 sections be changed? Also, were all extracted studies performed in the same way as suggested in 2.1? If not, it needs to be clarified on this point.

3) Materials and Methods: All of the selected studies were retrospective design. In the Material and Methods, it is necessary to describe the range of evidence level in this selected criterion (such as from RCT to retrospective study) and the reason for including the retrospective design should be explained (ex. this surgical method is still in early stages et al.), which enforced in strength and limitations sections.

4) Results: The assessment of biased risk analysis in the literature seems to be missing in this study.

5) Was there no difference in indication between the selected studies?

6) In discussion, it is essential to derive the advantages of disc-fx. Compared to other treatments through review of literatures in this study. In this review, it should not be just an introduction for Disc-fx..

7) The title should be modified considering the range of study : such as A scoping review / Systematic review and meta-analysis. As current form, it is not enough for systematic review and meta-analysis

Author Response

Reviewer 2

I thank the editor for giving me the opportunity to review your paper. Below, I have provided our comments.

Title

Effectiveness evaluation of nucleo-annuloplasty for lumbar discogenic lesions using Disc-FX: A systematic review

Critique

This systematic review deal with the effectiveness of nucleo-annuloplasty for lumbar discogenic lesions using Disc-FX. This system, as minimally invasive approaches, provide many advantages compared to other methods. Therefore, it has archival values but there have some concerns in this study.

1) Introduction: As far as I’m concerned (as reviewer), I think the indication of nucleo-annuloplasty is essential issues in this study. In the background, the indication must be clearly addressed. Where indication was overlapped and many studies have been suggested the effectiveness compared to other surgeries. The systematic reviews were recommended that this point be presented in the background.

  Thank you for your kind comments. In the introduction section we have written the indications for Disc-FX, which are degenerative disk disease or lumbar disk herniation resulting in diskogenic pain and contained type herniated nucleus pulposus.

2) Materials and Methods: Shouldn’t the order of 2.1 and 2.2 sections be changed? Also, were all extracted studies performed in the same way as suggested in 2.1? If not, it needs to be clarified on this point.

  Thank you for your kind comments. The order of 2.1 and 2.2 was changed according to your comments; in addition, we extracted the data according to the method described in 2.1.

3) Materials and Methods: All of the selected studies were retrospective design. In the Material and Methods, it is necessary to describe the range of evidence level in this selected criterion (such as from RCT to retrospective study) and the reason for including the retrospective design should be explained (ex. this surgical method is still in early stages et al.), which enforced in strength and limitations sections.

  Thank you for your kind comments. In accordance with your comments, relevant content has been added to section 2.2.

4) Results: The assessment of biased risk analysis in the literature seems to be missing in this study.

 Thank you for your kind comments. We added a quality assessment of the included studies as Table 1.

5) Was there no difference in indication between the selected studies?

  Thank you for your kind comments. No, it's all pretty much the same. The indications for all papers are degenerative disk disease or lumbar disk herniation resulting in diskogenic pain and contained type herniated nucleus pulposus.

6) In discussion, it is essential to derive the advantages of disc-fx. Compared to other treatments through review of literatures in this study. In this review, it should not be just an introduction for Disc-fx.

 Thank you for your kind comments. As per your suggestion, the advantages of Disc-FX have been added to the discussion section 4.2.

7) The title should be modified considering the range of study: such as A scoping review / Systematic review and meta-analysis. As current form, it is not enough for systematic review and meta-analysis

 Thank you for your kind comments. We have changed the title according to your comments.
